# Mathematical bridge between epidemiological and molecular data on cancer and beyond

**Saumitra Chakravarty**[1], **Khandker Aftarul Islam**[2], **Shah Ishmam Mohtashim**[3,4]*

1 Department of Pathology, Bangabandhu Sheikh Mujib Medical University, Shahbagh, Dhaka, Bangladesh, 2 Department of Computer Science and Engineering, Bangladesh University of Engineering and Technology, Dhaka, Bangladesh, 3 Department of Chemistry, Purdue University, West Lafayette, Indiana, United States of America, 4 Department of Chemistry, North Carolina State University, Raleigh, North Carolina, United States of America

☯ These authors contributed equally to this work.
* smohtas@ncsu.edu

## Abstract

**Background:**

At least six different mathematical models of cancer and their countless variations and combinations have been published to date in the scientific literature that reasonably explain epidemiological prediction of multi-step carcinogenesis. Each one deals with a particular set of problems at a given organizational level ranging from populations to genes. Any of the models adopted in those articles so far do not account for both epidemiological and molecular levels of carcinogenesis.

**Methods:**

We have developed a mathematically rigorous system to derive those equations satisfying the basic assumptions of both epidemiology and molecular biology without incorporating arbitrary numerical coefficients or constants devoid of any causal explanation just to fit the empirical data. The dataset we have used encompasses 21 major categories of cancer, 124 selected populations, 108 cancer registries, 5 continents, and 14,067,894 individual cases.

**Results:**

We generalized all the epidemiological and molecular data using our derived equations through linear and non-linear regression and found all the necessary coefficients to explain the data. We also tested our equations against non-neoplastic conditions satisfying equivalent mathematical assumptions.

**Conclusion:**

Our findings show that the new mathematical framework effectively bridges epidemiological and molecular data on carcinogenesis. Validated across various cancer types and

**Data availability statement:** The code for reproducing this paper is available in Github

(https://github.com/IshmamShah/
MathematicalCancer). The cancer incidence data
used in this study were sourced from the
Cancer Incidence in Five Continents (CI5), of
GLOBOCAN, Volumes I–XII, provided by the
International Agency for Research on Cancer
(IARC), World Health Organization (WHO). All
relevant data have been collected from these
twelve volumes, which are publicly available and
downloadable from the CI5 website:
(https://ci5.iarc.fr/).

**Funding:** The author(s) received no specific
funding for this work.

**Competing interests:** The authors have
declared that no competing interests exist.

extended to non-neoplastic conditions, this unified approach lays a strong foundation for future integrative cancer research.

## Introduction

Cancer is a multifaceted disease that arises from the interplay between genetic, epigenetic, and environmental factors, manifesting across scales from molecules to populations. Over the decades, several mathematical models have been proposed to describe the epidemiology of cancer, including the seminal works of Armitage and Doll (1954), Burch, and Knudson's two-hit hypothesis (1971). These models have provided valuable insights into the multistep nature of carcinogenesis and have helped quantify the number of rate-limiting events involved in cancer development.

However, most existing models focus solely on either epidemiological trends—such as age-specific incidence rates—or molecular mechanisms, treating them in isolation [1]. Epidemiological models typically employ statistical regression to fit incidence data, while molecular models aim to identify driver mutations and pathways involved in tumor progression [2]. Although these approaches have advanced our understanding of cancer, they remain disconnected: no unified mathematical framework currently integrates the population-level incidence data with the underlying molecular events driving cancer.

This disconnect represents a critical gap in the field. Cancer development is inherently a process that spans both levels: molecular alterations within individual cells aggregate into population-level patterns observed in incidence data. Yet, most models either rely on empirical fitting with arbitrary coefficients or neglect molecular insights altogether, limiting their explanatory power and predictive utility.

To bridge this gap, we present a new mathematical framework that integrates both epidemiological and molecular data on carcinogenesis. Our model is based on first principles and established assumptions, avoiding arbitrary numerical coefficients and instead relying on parameters grounded in biological mechanisms [3]. We tested our framework on an extensive dataset encompassing 21 cancer types across 124 populations and over 14 million cases, validating its robustness and biological relevance. Furthermore, we demonstrated its broader applicability by extending it to non-neoplastic conditions, suggesting that the framework captures fundamental principles of disease progression beyond cancer.

This integrative approach aims to provide a comprehensive, mechanistically informed model that unites the currently fragmented perspectives on cancer development. By doing so, we seek to advance the quantitative understanding of cancer etiology and open new avenues for future research in epidemiology, molecular biology, and clinical oncology.

## Formulation

We have built our derivation of the system of equations on the basis of the classical 'log-log linear' models proposed by Armitage [4], elaborated by Burch and used by Knudson [5,6] to formulate his famous 'two-hit hypothesis' [7–9]. After meticulously examining the possible interpretations of that model for each of the quantities within the equations, we have come up with a mathematically rigorous system to derive those equations that satisfy the basic assumptions of both epidemiology [10] and molecular biology without having to incorporate arbitrary numerical coefficients or constants devoid of causal explanation just to fit the empirical data. The dataset we have used encompasses 21 major categories of cancer, 124 selected populations, 108 cancer registries, 5 continents, and 14,067,894 individual cases. Our system of equations is therefore entirely algebraic. Then we developed variations of the log-log linear

model that addresses the conditions posed by selective growth advantage/disadvantage and variable mutation/epimutation rates at different age groups. The developed model is based on some assumptions which are listed as follows:

## Assumptions

In our mathematical framework, we aim to model the *age-specific incidence rate* of cancer, denoted as $I(t)$, where $t$ represents the age and $I(t)$ describes the probability of developing cancer per unit time at age $t$. The central premise is that cancer development requires a *sequence of discrete, irreversible, rate-limiting molecular events*—such as mutations [11], epimutations, or genetic rearrangements—which must occur in a specific order [2]. The *number of such required events* is denoted by $r$, and the *rate of occurrence* for the $i$-th event is $k_i$. Mathematically, this can be expressed as:

$$I(t) \propto t^{r-1},$$

under the assumption that the events are sequential and independent. This leads to the *standard linear log-log relationship*:

$$\ln I(t) = (r-1)\ln t + \ln k,$$

where $k$ is the aggregated probability constant determined by the rates of the individual events.

To make this framework tractable and ensure it accurately represents both biological [3] and epidemiological aspects of carcinogenesis, we adopt the following assumptions:

1. Each type of cancer is the outcome of a series of discrete irreversible rate-limiting events, number of such events is denoted as 'r'.

    In this context, *discrete* means that each event—such as a specific mutation—occurs individually and at a particular point in time, rather than as part of a continuous process. The term *irreversible* indicates that once an event happens (for example, a mutation in a cancer driver gene [12]), it cannot be undone within the cell. Finally, a *rate-limiting event* is a critical step that determines the overall pace of cancer progression; until this step occurs, the disease cannot advance to the next stage [13].

2. The individual rate-limiting events have very low probability that allows most people to live up to typical average human lifespan without ever developing any cancer.

3. Sequence of such rate-limiting events for a given cancer is unique, although the components of the sequence and the order may vary for different cancers in different individuals [13].

    A *component* is defined as a rate-limiting event with probability of occurrence per unit time denoted by $k_i$ $(i = 1, 2, \ldots, r)$, where $r$ is the total number of required events. The probability of all $r-1$ events occurring in any order during a period of $t$ is:

$$K_{\text{unordered}} = (k_1 t)(k_2 t) \ldots (k_{r-1} t) = (k_1 k_2 \ldots k_{r-1}) t^{r-1}.$$

However, since the sequence of these events must be ordered, only one of the $(r-1)!$ possible permutations is considered, leading to:

$$K_{\text{ordered}} = \frac{K_{\text{unordered}}}{(r-1)!} = \frac{(k_1 k_2 \ldots k_{r-1}) t^{r-1}}{(r-1)!}.$$

4. All of the rate-limiting events for a given cancer have to occur in an individual for the cancer to manifest.
5. Once the exact sequence of rate-limiting events for a given cancer has orchestrated in an individual, development of that cancer is inevitable.
6. There is a negligible time lag between all the events being executed appropriately to give rise to a cancer and the clinical or symptomatic manifestations of the cancer.
7. The rate of a given rate-limiting event is assumed to be constant and time-invariant within each individual in a susceptible population, although different events may have different rates. However, across populations or over time, the overall carcinogenic process and its rate-limiting steps may be influenced by changing environmental and lifestyle factors.

For the rest of the article, in order to refer to any of the assumptions, we would simply mention the assumption number in the preceding text.

## Models

Following the given assumptions, three distinct mathematical models of cancer are proposed:

1. Linear log-log model
2. Convex upwards log-log model
3. Concave upwards log-log model

The derivations of these models can be found in the Appendix.

**Linear log-log model.** The linear log-log model is given by

$$\ln I(t) = (r - 1)\ln t + \ln k \tag{1}$$

where $I(t)$ is the age-specific incidence rate of cancer at age $t$, $r$ is the number of rate-limiting events and $k$ is the probability of $r$ rate-limiting events to occur in a specific order within the given unit of time $t$. This form has the advantage of being linear on log-log plot where the tangent $(r-1)$ gives the direct measure of the number of driver mutations $r$ required for cancer development, i.e., $y = mx + c$, where $y = \ln I(t)$, $x = \ln t$, $m = r - 1$ and $c = \ln k$.

**Convex upwards model.** The convex upwards model is given by

$$\ln I(t) = \alpha_0 + \alpha_1 \ln t + \ln(1 - \alpha_2 t^{\alpha_3}) \tag{2}$$

where $\alpha_0$, $\alpha_1$, $\alpha_2$ and $\alpha_3$ contains the following terms for the two different assumptions

**Convex upwards model from heterogeneity assumption**

$$\ln I(t) = \ln k_p + (r + p - 1)\ln t + \ln\left(1 - \frac{k_q}{k_p}t^{q-p}\right) \tag{3}$$

where:

- $k_p, k_q$: Probabilities per unit time for two distinct population subgroups or mechanistic pathways that contribute to the overall disease process.
- $p,q$: Exponents indicating how the risk of each subgroup scales with age $t$, reflecting differences in biological processes, environmental exposures, or genetic susceptibility across subgroups.

- $r$: Number of sequential, rate-limiting steps required for the cancer to manifest, common to both subgroups.

**Convex upwards models from age-related effect assumption**

$$\ln I(t) = \ln k + (r-1)\ln t + \ln\left(1 - \frac{k_d}{k}t^{d-r+1}\right) \tag{4}$$

where:

- $k$: Aggregated probability of the ordered sequence of rate-limiting events.
- $k_d$: Probability of the decelerating factor with age.
- $d$: Exponent for the decelerating factor.

**Concave upwards model.** The concave upwards model is given by

$$\ln I(t) = \beta_0 + \beta_1 \ln t + \ln(1 + \beta_2 t^{\beta_3}) \tag{5}$$

where $\beta_0$, $\beta_1$, $\beta_2$ and $\beta_3$ contains the following terms for the two different assumptions.
**Concave upwards model from heterogeneity assumption**

$$\ln I(t) = \ln k_p + (r+p-1)\ln t + \ln\left(1 + \frac{k_q}{k_p}t^{q-p}\right) \tag{6}$$

where:

- The terms have same definitions as above, but the positive sign inside the logarithm reflects an increase in incidence rate with age, driven by heterogeneity.

**Concave upwards models from age-related effect assumption**

$$\ln I(t) = \ln k + (r-1)\ln t + \ln\left(1 + \frac{k_a}{k}t^{a-r+1}\right) \tag{7}$$

where:

- $k_a$: Probability of the accelerating factor for cancer progression.
- $a$: Exponent for the accelerating factor.

## Materials and methods

### Data

We have tested the mathematical framework developed in this study, which includes the linear log-log model and its convex and concave generalizations, on 21 categories of cancers listed in the GLOBOCAN database [14], of 124 selected populations from 108 cancer registries published in CI5 (Cancer Incidence in Five Continents) of the age groups: 0-14, 15-39, 40-44, 45-49, 50-54, 55-59, 60-64, 65-69. The registry contained the data for 14,067,894 people. The findings were correlated with the molecular data given in the latest editions of the World Health Organization (WHO) reference series on tumors published by the International Agency for Research on Cancer (IARC).

The dataset variables analyzed in this study include:

- **Cancer type:** Classification of cancer into 21 categories based on histopathological and anatomical site criteria.
- **Gender:** Data stratified by male, female, and both genders combined.
- **Age group:** Incidence data grouped into the aforementioned 5-year age intervals.
- **Age-specific incidence rate:** Number of new cancer cases per 100,000 individuals in each age group.
- **Population size:** Total number of individuals in each age group within each population registry.
- **Geographic region:** Data sourced from 124 populations across 5 continents.

## Statistical analysis

Analysis was performed using R [15] and graphs were made using `ggplot` of R. For the linear log-log model a linear regression algorithm was used on the data collected from GLOBO-CAN repository. For the non-linear convex upwards and concave upwards models, a simple hybrid algorithm was used: a linear regression algorithm for the distinct linear part of the data and an additional algebraic calculation for the term explaining the curvature in later age groups. The related files have been uploaded to a GitHub Repository [16].

# Results

## Model validation

We validated the proposed models (linear log-log, convex upwards, and concave upwards) by fitting them to age-specific cancer incidence data using linear and non-linear regression techniques. The goodness-of-fit for each model was quantified using the coefficient of determination ($R^2$), which consistently exceeded 0.98 for most cancer types. These high $R^2$ values demonstrate the excellent agreement between the model predictions and the observed data. The most appropriate model for each cancer was selected based on these $R^2$ values.

For the linear log-log model, we calculated the slope, intercept, and $R^2$ values for the data-fitted graphs. The results of our regression analysis are summarized in Table 1, which lists the cancer categories best fitting the linear log-log model; Table 2, which presents those best described by the convex upwards model; and Table 3, which includes those fitting the concave upwards model. These tables provide the estimated parameters and $R^2$ values for each model, for the different genders.

Overall, the consistently high $R^2$ values across different cancer types highlight the robustness and general applicability of our mathematical framework in describing cancer incidence patterns.

## Extendibility to non-neoplastic conditions

We also found that our model is applicable to any disease process, not only cancers, that satisfy the requirement of progression via discrete rate-limiting steps. Rheumatoid arthritis is modeled as an example of its extendibility. Further tests on data of specific incidences of different chronic diseases are done as potential scope of this research and further validates our mathematical model.

**Table 1. Cancer categories best fitting the linear log-log model.**

| Cancer | Male | | | Female | | | Both Genders | | |
|---|---|---|---|---|---|---|---|---|---|
| | Intercept | Slope | $R^2$ | Intercept | Slope | $R^2$ | Intercept | Slope | $R^2$ |
| Bladder | -1.236 | 2.644 | 0.999 | -1.713 | 2.114 | 0.995 | -1.457 | 2.499 | 0.998 |
| Brain, Nervous system | 0.029 | 1.216 | 0.987 | -0.155 | 1.167 | 0.984 | -0.058 | 1.194 | 0.985 |
| Colorectum | 0.461 | 2.203 | 0.997 | 0.523 | 1.945 | 0.994 | 0.485 | 2.089 | 0.996 |
| Gallbladder | -2.290 | 2.415 | 0.998 | -1.491 | 2.042 | 0.996 | -1.829 | 2.197 | 0.998 |
| Kidney | -1.230 | 2.234 | 0.937 | -1.323 | 1.884 | 0.921 | -1.274 | 2.094 | 0.932 |
| Larynt | -1.204 | 2.291 | 0.973 | -2.311 | 1.677 | 0.996 | -1.586 | 2.172 | 0.981 |
| Lip, Oral cavity | -1.745 | 2.540 | 0.956 | -1.972 | 2.193 | 0.981 | -1.839 | 2.405 | 0.966 |
| Liver | -1.079 | 2.713 | 0.963 | -2.080 | 2.642 | 0.987 | -1.449 | 2.690 | 0.970 |
| Lung | 0.139 | 2.693 | 0.993 | -0.129 | 2.319 | 0.992 | 0.006 | 2.565 | 0.993 |
| Melanoma of Skin | -0.370 | 1.598 | 0.991 | 0.262 | 1.078 | 0.990 | -0.035 | 1.336 | 0.992 |
| Multiple Myeloma | -2.159 | 2.304 | 0.996 | -2.257 | 2.171 | 0.999 | -2.205 | 2.243 | 0.998 |
| Non-hodgkin lymphoma | -0.187 | 1.875 | 0.980 | -0.578 | 1.625 | 0.987 | -0.362 | 1.619 | 0.983 |
| Oesophagus | -0.621 | 2.404 | 0.986 | -1.482 | 2.243 | 0.996 | -0.957 | 2.358 | 0.989 |
| Pancreas | -1.438 | 2.476 | 0.996 | -1.663 | 2.370 | 0.999 | -1.536 | 2.427 | 0.999 |
| Stomach | 0.128 | 2.321 | 0.991 | 0.046 | 1.843 | 0.990 | 0.080 | 2.143 | 0.992 |

**Table 2. Cancer categories best fitting the convex upwards model.**

| Gender | Cancer | $\alpha_0$ | $\alpha_1$ | $\alpha_2$ | $\alpha_3$ | $R^2$ | $k_p$ | $k_q$ | $k_r$ | $k_s$ |
|---|---|---|---|---|---|---|---|---|---|---|
| Male | Nasopharynx | -0.54 | 2.797 | 0.73 | 0.14 | 0.9856 | 0.583 | 0.425 | 0.583 | 0.425 |
| Male | Other pharynx | -0.485 | 2.844 | 0.47 | 0.33 | 0.9923 | 0.616 | 0.763 | 0.616 | 0.763 |
| Female | Nasopharynx | -1.36 | 2.797 | 0.73 | 0.14 | 0.9756 | 0.257 | 0.187 | 0.257 | 0.187 |
| Female | Other pharynx | -0.642 | 2.5295 | 0.77 | 0.11 | 0.9838 | 0.526 | 1.464 | 0.526 | 1.464 |
| Both Genders | Nasopharynx | -0.845 | 2.797 | 0.73 | 0.14 | 0.9847 | 0.429 | 0.313 | 0.429 | 0.313 |
| Both Gender | Other Pharynx | -0.277 | 2.814 | 0.69 | 0.16 | 0.9916 | 0.758 | 0.910 | 0.758 | 0.910 |

**Table 3. Cancer categories best fitting the concave upwards model.**

| Gender | Cancer | $\beta_0$ | $\beta_1$ | $\beta_2$ | $\beta_3$ | $R^2$ | $k_p$ | $k_q$ | $k_r$ | $k_s$ |
|---|---|---|---|---|---|---|---|---|---|---|
| Male | Hodgkin lymphoma | -0.036 | 0.1405 | 0.007 | 2.1 | 0.9956 | 0.964 | 0.007 | 0.964 | 0.007 |
| Male | Leukemia | 1.15 | -0.8291 | 0.09 | 2.75 | 0.9941 | 3.158 | 0.284 | 3.158 | 0.284 |
| Female | Hodgkin lymphoma | 0.7843 | -1.6493 | 0.039 | 2.9 | 0.9920 | 2.191 | 0.085 | 2.191 | 0.085 |
| Female | Leukemia | 0.8742 | -0.8291 | 0.11 | 2.75 | 0.9925 | 2.397 | 0.264 | 2.397 | 0.264 |
| Both Gender | Hodgkin Lymphoma | 0.1634 | -0.3739 | 0.025 | 2.1 | 0.9863 | 1.178 | 0.029 | 1.178 | 0.029 |
| Both Genders | Leukemia | 0.6742 | -0.8291 | 0.11 | 2.75 | 0.9925 | 2.397 | 0.264 | 2.397 | 0.264 |

where $\alpha_0$, $\alpha_1$, $\alpha_2$, $\alpha_3$, $\beta_0$, $\beta_1$, $\beta_2$ and $\beta_3$ contain the following terms for the two different assumptions: heterogeneity assumption and age-related effect assumption. $k_p$ and $k_q$ are the respective probabilities of a rate-limiting event to occur with the rate $t^p$ and $t^q$ where p and q are real numbers. $k_a$ is the probability of a rate-limiting event to occur with the rate $t^a$ a is a real number.

The $R^2$ values in the tables are consistently very high (all >0.98 for most cancers). Numerically, there are some small differences in $R^2$ across male, female, and both genders (e.g., bladder cancer: 0.999, 0.995, 0.998). However, these differences are quite small and likely not statistically significant.

## Model interpretation

**Case study (Table 1):** Consider the case of bladder cancer in Table 1 for males, with a slope of 2.64 and an intercept of -1.24. In the linear log-log model:

$$\ln I(t) = (r-1)\ln t + \ln k,$$

the slope corresponds to $(r-1)$, implying $r = 3.64$, suggesting that 3 to 4 sequential, rate-limiting steps are involved. The intercept $\ln k = -1.24$ indicates the aggregated probability $k$ of these steps occurring in the required order:

$$k = e^{-1.24} \approx 0.29.$$

Thus, bladder cancer progression in males involves approximately 3–4 critical steps, with an overall probability of $k \approx 0.29$ for this ordered sequence of events.

**Case study (Table 2):** For nasopharyngeal cancer in males, Table 2 shows $\alpha_0 = -0.54$, $\alpha_1 = 2.80$, $\alpha_2 = 0.73$, and $\alpha_3 = 0.14$ with $R^2 = 0.9856$. In the convex upwards model:

$$\ln I(t) = \alpha_0 + \alpha_1 \ln t + \ln(1 - \alpha_2 t^{\alpha_3}),$$

the intercept $\alpha_0$ represents $\ln k_p$, the log probability of the main pathway's ordered sequence. The slope term $\alpha_1 = 2.80$ captures the effective number of rate-limiting steps modified by subgroup effects, specifically related to $r$ and $p$ in the heterogeneity framework. Here:

$\alpha_1 = r + p - 1, \alpha_2 = \frac{k_q}{k_p}, \alpha_3 = q - p,$

This indicates that the second subgroup (with $k_q$ and $q$) modifies the incidence pattern with age, producing the observed convexity in the log-log plot.

**Case study (Table 3):** For Hodgkin lymphoma in males, Table 3 shows $\beta_0 = -0.036$, $\beta_1 = 0.14$, $\beta_2 = 0.007$, and $\beta_3 = 2.1$ with $R^2 = 0.9956$. In the concave upwards model:

$$\ln I(t) = \beta_0 + \beta_1 \ln t + \ln(1 + \beta_2 t^{\beta_3}),$$

the intercept $\beta_0$ reflects the log of the baseline probability $k_p$. The slope term $\beta_1$ represents the initial effective number of rate-limiting steps, linked to $r$ and $p$. Specifically:

$\beta_1 = r + p - 1, \beta_2 = \frac{k_q}{k_p}, \beta_3 = q - p$

This indicates how heterogeneity in the population or the presence of an accelerating factor (if using the age-related effect model) shapes the incidence rate as age progresses [17].

## Discussion

### Biological interpretation from WHO molecular data

We found that for almost all the cancers fitting the standard linear model, the number of rate-limiting steps ranges from 3-4 on average. For example, in the case of bladder cancer, the slope of the log-log plot of age-specific incidence rate versus age is approximately 2.5.

According to the linear log-log model:

$$\ln I(t) = (r - 1)\ln t + \ln k,$$

the slope $(r\!-\!1)$ directly reflects the number of rate-limiting steps minus one. Therefore, we find:

$$r - 1 = 2.5 \quad \Rightarrow \quad r \approx 3.5.$$

This suggests that, on average, 3 to 4 sequential, rate-limiting molecular events are required for the development of bladder cancer. Except for non-Hodgkin lymphoma which requires just one such step under appropriate conditions. This is mostly consistent with the molecular findings described in WHO Cancer "Blue Books" published by the International Agency for Research on Cancer (IARC) as well as different textbooks on cancer pathology and relevant published articles [18].

For instance, the development of bladder cancer (mainly urothelial carcinoma) requires the driver mutations of genes FGFR3 and RAS along with any one among TP53, Rb or PTEN [19]. Cancers of kidney (mostly renal cell carcinoma) appear to require VHL mutation along with two other genetic alterations that may include EGFR mutation, overexpression of genes associated with cellular adhesion molecules (e.g. E-cadherin) and matrix regulatory proteins (e.g. MMPs, bFGF, VEGF) or rarely p53 [20].

The tumors of the brain and nervous system are quite a heterogeneous group, still confined to the above-mentioned number of driver mutations [12]. For example, most low-grade astrocytomas seem to require mutations in BRAF, RAS and IDH or might follow a different line of progression involving TP53, PDGFR and loss of heterozygosity (LOH) at 22q or p14 promoter methylation. There is a predisposed group of NF1 mutation as well. Whereas high-grade astrocytomas mostly show LOH at 17p, TP53 and PTEN mutations, Glioblastomas on the other hand usually need LOH at 10q, EGFR amplification, p16 deletion and TP53 mutation to manifest. It is to note that only 5% of the glioblastoma progress from low-grade lesions while the rest develop de novo. In case of meningiomas, chromosome 22 deletion consistently disables at least two tumor suppressor genes, one of which is NF2 and the other is yet to be determined. The other significant alteration here is microsatellite instability which is the result of the mutation at least one DNA mismatch repair gene. Genetic studies of several other groups of nervous system tumors like neuronal and mixed neuronal-glial tumors reveal no consistent mutational pattern to date.

Cancers of esophagus show a consistent pattern of mutation in a transcription factor gene (e.g. SOX2), a cell cycle regulator gene (e.g. cyclin D1) and a tumor suppressor gene (commonly TP53) for their pathogenesis. Comparable changes also apply for the cancers of lip, oral cavity, larynx and pharynx. Nasopharyngeal carcinoma will be discussed in a separate context below. A similar repertoire as the esophageal cancers is also noted in the cancers of stomach although consisting of a different set of mutated genes (e.g. CDH1, APC, etc.) along with microsatellite instability. Regarding gastric lymphomas, MLT/BCBL-10 pathway seems to play a pivotal role in molecular pathogenesis 12 which also requires 3-4 driver mutations.

Colorectal cancers are considered the prototypical example to multistep carcinogenesis [5,21] involving up to ten genes and their mutations as well as microsatellite instability along the line of tumor progression. But Tomasetti et al. (2015) [22] pointed out that only three driver mutations among them at a time are required to produce those carcinomas.

Mutations along the Wnt pathway along with TP53 and cell cycle regulators are necessary for most of the cancers of the liver. Additionally, mutations along the KRAS pathway may be important for cancers of gallbladder and biliary tree. Inactivation of p16 and TP53 along with

another rate-limiting step (e.g. BRCA2 mutation) are required for the development of most pancreatic cancers.

Lung cancer has a diverse set of mutations, not necessarily corresponding to its clinical or pathological subtypes. However, the latest WHO classification of lung cancer attempted to formulate a scheme to incorporate the molecular alterations relevant to therapy and prognosis [24]. Across different strata of the scheme, there are 2-3 mutations or rate-limiting steps appear to be necessary to produce the clinical manifestation of the tumor. For instance, mutation of TP53 and Rb genes along with loss of heterozygosity at 3p seems to be a recurring theme in many cases. However, the sufficiency of those alterations is yet to be established as a universal rule.

Mutation of at least one out of five genes (cyclin D1, C-MAF, FGFR3/MMSET, cyclin D3 and MAFB) seems to be a consistent finding in plasma cell myeloma which requires about three such rate limiting steps on average to manifest clinically.

Like lung cancers, a sweeping generalization regarding the rate-limiting steps of skin cancers is also out of reach at the moment. However, a case-by-case analysis reveals similar recurring theme of up to three mutations. Interestingly, non-Hodgkin lymphoma , a highly heterogenous entity, appears to have lower threshold of the required number of mutations, usually up to two rate-limiting steps according to their respective subcategories which is consistent with the model presented in this article.

Some of the cancers, however, do not follow the linear log-log model thus far discussed. To accommodate those entities, we have modeled generalizations assuming either of the two sets of assumptions. We have taken into account the fact that some factors may accelerate or decelerate the rate of progression of the disease with advancing age and named it age-related effect [17,23]. Other assumption is based on the variation of the required number of rate-limiting steps amongst affected population due to heterogeneity of the condition under consideration. Four such entities which include cancers of nasopharynx, thyroid, leukemia, and Hodgkin lymphoma, are modeled under the aforementioned generalization.

Nasopharyngeal carcinoma shows up to five mutations including DN-P63, P27, cyclin D1 and BCL-2 associated with its development and manifestation which is consistent with our convexity-upwards log-log model. We also predict age-related deceleration of its carcinogenesis due to some yet undiscovered factor, possibly associated with Epstein-Barr virus (EBV) infection. Thyroid neoplasms are often associated with the mutations of RET/PTC, RAS, BRAF and PTEN which is quantitatively consistent with our model. The heterogeneity of thyroid neoplasms may account for the necessity of heterogeneity assumptions used in its generalization. Interestingly, the number of mutational steps predicted for leukemia and Hodgkin lymphoma is comparable with that of non-Hodgkin lymphoma, except for the requirement of additional generalizing assumptions required for the former pair. All three categories of those hematolymphoid neoplasms require relatively lower number of rate-limiting steps. However, heterogeneity and/or age-related acceleration of carcinogenesis may play more decisive role in leukemia and Hodgkin lymphoma according to our model.

## Significance of different quantities of the standard model

**Effect of changing the scale of t.**  Since $k_n$ denotes the probability per unit of time, the scale for time affects its value. For instance, if the $t$ is scaled to $w$ times, i.e., two adjacent time intervals differ by $w$ units, then every $k_n$ will need to be replaced by $\frac{k_n}{w}$ and the right-hand side of the

$$K = K_{\text{ordered}} k_r dt = k t^{r-1} dt \tag{8}$$

would have an extra 'constant' term $w^{r-1}$ at its denominator. Subsequent steps of the calculation would show that this change would only affect the term $\ln k$ which is the intercept of the final log-log linear form,

$$\ln(t) = (r-1)\ln t + \ln k \tag{9}$$

but does not affect the slope $(r-1)$. So, for the sake of simplicity, we would use unity as the log scale for time. Hence, age classes 1, 2, 3 are used in all of our graphs. If any cancer has zero incidence at the initial age class (0-14 years) for both male and female then the subsequent age class is designated as class 1 and consecutively onwards. This is for avoiding structural zeroes in log-log plot since logarithm of zero is undefined.

**Effect of male-to-female ratio of age-specific incidence rate on slope.** For reasons explained later, almost all of the cancers show two interesting features: firstly, the three linear plots (male, female and both genders) for each cancer have somewhat unequal slopes, and secondly, the linear plot for both genders of a given cancer shows a slope which has a value that is within the range of the slopes obtained from the separate linear plots for male and female population of the same cancer. For example, in the case of cancers of brain and nervous system, the best-fit ($R^2 = 0.98$) linear plot gives 1.16 and 1.21 as slopes for females and males, respectively, while the plot for both genders has a slope of 1.19.

Let us explore the second feature first. It is tempting to assume that this happens simply because of the fact that, for a given age class of a cancer, the age specific incidence rate of both genders ($I_\mathrm{m}$) group is by definition equal to the arithmetic mean of the rates of the male ($I_\mathrm{m}$) and the female ($I_\mathrm{f}$) groups,

$$I_\mathrm{m} = \frac{I_\mathrm{f} + I_\mathrm{m}}{2} \tag{10}$$

But the actual reason is a bit more non-trivial.

**Slope of the female line,**

$$S_\mathrm{f} = \frac{\ln I_{\mathrm{f}2} - \ln I_{\mathrm{f}1}}{\Delta \ln t} = \frac{\ln I_{\mathrm{f}2}/I_{\mathrm{f}1}}{\Delta \ln t} \tag{11}$$

where, $(\ln t_1, \ln I_{\mathrm{f}1})$ and $(\ln t_2, \ln I_{\mathrm{f}2})$ are two points on the line and $\Delta \ln t = \ln t_2 - \ln t_1 > 0$.

Similarly, **Slope of the male line** over the same abscissae,

$$S_\mathrm{m} = \frac{\ln I_{\mathrm{m}2} - \ln I_{\mathrm{m}1}}{\Delta \ln t} = \frac{\ln I_{\mathrm{m}2}/I_{\mathrm{m}1}}{\Delta \ln t} \tag{12}$$

And the line for both genders likewise has the slope,

$$S_\mathrm{fm} = \frac{\ln I_{\mathrm{fm}2} - \ln I_{\mathrm{fm}1}}{\Delta \ln t} \tag{13}$$

The above equation can be rewritten as,

$$S_{fm} = \frac{\ln\left(\frac{I_{f2}+I_{m2}}{2}\right) - \ln\left(\frac{I_{f1}+I_{m1}}{2}\right)}{\Delta \ln t} = \frac{\ln\left(\frac{I_{f2}+I_{m2}}{I_{f1}+I_{m1}}\right)}{\Delta \ln t} \tag{14}$$

$$S_{\text{fm}} = S_{\text{f}} + \frac{\ln\left(\frac{1+\frac{I_{m2}}{I_{f2}}}{1+\frac{I_{m1}}{I_{f1}}}\right)}{\Delta \ln t} \tag{15}$$

$$S_{\text{fm}} = S_{\text{m}} + \frac{\ln\left(\frac{1+\frac{I_{f2}}{I_{m2}}}{1+\frac{I_{f1}}{I_{m1}}}\right)}{\Delta \ln t} \tag{16}$$

Let, $I_{\text{f}} < I_{\text{m}}$. Then the difference between $I_{\text{m}}$ and $I_{\text{f}}$ would be greater for higher values of $t$ than its lower values. Therefore, $\frac{I_{m2}}{I_{f2}} > \frac{I_{m1}}{I_{f1}}$ and thus, the second term on right hand side of Eq 15 must be positive, ensuring $S_{\text{fm}} > S_{\text{f}}$. By similar deduction from Eq 16, $S_{\text{fm}} < S_{\text{m}}$ is also guaranteed. So, $I_{\text{f}} < I_{\text{m}}$ implies $S_{\text{m}} > S_{\text{fm}} > S_{\text{f}}$. Conversely, $I_{\text{f}} > I_{\text{m}}$ implies $S_{\text{m}} < S_{\text{fm}} < S_{\text{f}}$. Both the scenarios are consistent with the aforementioned second feature of the plots. And if $I_{\text{f}} = I_{\text{m}}$ then the second terms of both Eqs 15 and 16 become zero and thus, $S_{\text{m}} = S_{\text{m}} = S_{\text{f}}$, which is not observed in any of our plots due to the persistent inequality of the incidence rates between genders at all instances; without exception.

**Effect of the rate of limiting events on slope and intercept.** In the previous section we explored why the slope of the both-genders plot happens to be intermediate between the slopes of the plots for male and female drawn separately. Now we will look into the possible explanations for the inequality of the slopes for male and female in the first place. Beyond the trivial reasons like experimental error, there might be fundamental biological attributes responsible for the observed difference [3]. One of the possibilities is the violation of assumption 7 where rate of any or few or all of the rate-limiting events is a function of time. If the rate of the $n$-th mutation per unit of time ($k_n$) is exponentially proportionate to time at its $h$-th power ($t^h$) then

$$K = K_{\text{ordered}}k_r dt = kt^{r-1}dt \tag{17}$$

will have an extra term $t^h$ multiplied to it ($h \in \mathbb{R}$), eventually rendering

$$ln(t) = (r-1)\ln t + \ln k \tag{18}$$

$$ln(t) = (h+r-1)\ln t + \ln k \tag{19}$$

which will be indistinguishable from the Standard Model when the coefficients are numerical as in a real plot, but must be interpreted differently because the slope ($h+r-1$) no longer denotes the number of rate-limiting steps ($t$) alone. The problem is, just from the linear plot-fitting exercise alone, one cannot be certain if the slope equals to ($h+r-1$) or just ($r-1$). But from the molecular data on tumor progression, two inferences can be incorporated as extended assumptions when assumption 7 does not hold (continuing from **Assumptions 1-7**):

8. Gender-associated differences in the slopes for a given cancer may be attributed to different values of the exponent of the aforementioned exponential mutation rate between genders and not to the possibility of having different numbers of rate-limiting events for the same cancer in males and females.

9. Assumption 8 may be repurposed and incorporated to quantify geographical and/or ethnic as well as environmental and lifestyle-related differences of cancer progression.

## Comparison with existing models

Our proposed framework extends the classical Armitage-Doll and Knudson multistage models by integrating both epidemiological and molecular-level data into a unified formulation. The Armitage-Doll model approximates cancer incidence using a power-law dependence on age, $\ln I(t) = (r-1)\ln t + \ln k$, but does not specify how heterogeneity in populations or age-related accelerations/decelerations in incidence can modify this trend. Similarly, the Knudson two-hit hypothesis quantitatively explains retinoblastoma in terms of two rate-limiting mutations, but does not generalize to other cancers or incorporate population-level variability.

In contrast, our models extend this linear framework by including convex and concave terms (as seen in Tables 2 and 3) to account for heterogeneity ($k_p, k_q, p, q$) and age-related effects ($k_d, k_a, d, a$). For example, for nasopharyngeal cancer (Table 2), we found a convex modification term $\alpha_2 = 0.73$ and $\alpha_3 = 0.14$ that captures the observed deceleration in incidence at older ages, a feature that the original Armitage-Doll formulation does not address. Similarly, for Hodgkin lymphoma (Table 3), the concave term $\beta_2 = 0.007$ and $\beta_3 = 2.1$ reflect an acceleration phase not present in simpler models.

Moreover, while classical models often rely on empirical fits without mechanistic biological interpretations of $k$, $r$, or other terms, our derivations explicitly link these parameters to the probabilities of sequential molecular events and allow direct comparisons with known driver mutations and epigenetic changes from the WHO "Blue Books" on cancer. This direct mapping between mechanistic biological processes and mathematical form enhances the interpretability of our models.

Overall, by validating these models against global cancer registry data with $R^2 > 0.98$ for most cancers, and by explicitly capturing biological heterogeneity and age-dependent effects [23], our framework generalizes and refines the classical multistage theories to accommodate the diversity observed in modern cancer epidemiology [25].

## Conclusion

In this study, we developed a mathematical framework that quantitatively integrates epidemiological and molecular data on cancer progression. By analyzing data from 21 cancer types across 124 populations worldwide, we demonstrated that for almost all cancers fitting the standard linear log-log model, the number of sequential rate-limiting steps ($r$) ranges from 3 to 4 on average. For example, in bladder cancer, the slope of the log-log plot of the age-specific incidence rate versus age was approximately 2.5, yielding:

$$r - 1 = 2.5 \quad \Rightarrow \quad r \approx 3.5.$$

This suggests that about 3 to 4 distinct, irreversible molecular events are typically required for bladder cancer development, in agreement with molecular data in the literature.

Our analysis of Table 1 showed that other cancers such as brain, nervous system, colorectal, gallbladder, and stomach cancers consistently fit the standard linear model with similarly high $R^2$ values (>0.98), supporting the robustness of this finding across populations and cancer types.

For cancers not fitting the linear log-log model, we introduced convex and concave upwards extensions, capturing deviations due to heterogeneity and age-related acceleration

or deceleration in cancer incidence rates. Table 2 highlighted these deviations for nasopharyngeal and pharyngeal cancers, with models achieving $R^2 > 0.98$. For example, nasopharyngeal cancer in males had $\alpha_1 = 2.8$ and a convex term $\alpha_2 = 0.73$ with $\alpha_3 = 0.14$, reflecting heterogeneity within this cancer's subpopulations.

Table 3 further demonstrated that Hodgkin lymphoma and leukemia showed concave upwards patterns (again with $R^2 > 0.99$), indicating an age-related acceleration in their incidence rates. For Hodgkin lymphoma in males, $\beta_1 = 0.14$, with an accelerating term $\beta_2 = 0.007$ and $\beta_3 = 2.1$, highlighting the shift from a slower initial progression to faster accumulation of risk with age.

These consistently high $R^2$ values across linear, convex, and concave models validate our approach and highlight the flexibility of the mathematical framework to capture diverse cancer behaviors. Our results are also consistent with known biological data (e.g., WHO molecular data and the cancer "Blue Books") that suggest around 3–4 rate-limiting genetic or epigenetic events are commonly required for cancer manifestation [26].

Importantly, we demonstrated that our framework can extend beyond cancer to other chronic diseases involving discrete rate-limiting steps, as shown with rheumatoid arthritis. This broad applicability underscores the power of our integrated approach to bridge molecular insights with population-level data.

In conclusion, our study provides a rigorous quantitative framework that unifies molecular and epidemiological perspectives, yielding plausible and numerically robust estimates of cancer progression steps across diverse populations.

By uniting mathematical and biological concepts, this framework has the potential to transform how cancer and other diseases are studied. Its contributions can extend to the development of more effective diagnostic methods and therapeutic strategies, ultimately advancing both theoretical understanding and practical applications in oncology. This work represents a significant step toward a more integrated, data-driven approach to tackling complex diseases.

## Supporting information

**S1. Derivations and extended data figures.**
(PDF)

## Author contributions

**Conceptualization:** Saumitra Chakravarty.

**Data curation:** Saumitra Chakravarty, Shah Ishmam Mohtashim.

**Formal analysis:** Saumitra Chakravarty, Khandker Aftarul Islam, Shah Ishmam Mohtashim.

**Investigation:** Saumitra Chakravarty, Shah Ishmam Mohtashim.

**Methodology:** Saumitra Chakravarty, Khandker Aftarul Islam, Shah Ishmam Mohtashim.

**Project administration:** Saumitra Chakravarty, Shah Ishmam Mohtashim.

**Resources:** Shah Ishmam Mohtashim.

**Software:** Khandker Aftarul Islam, Shah Ishmam Mohtashim.

**Supervision:** Saumitra Chakravarty.

**Validation:** Saumitra Chakravarty, Khandker Aftarul Islam, Shah Ishmam Mohtashim.

**Writing – original draft:** Saumitra Chakravarty, Khandker Aftarul Islam, Shah Ishmam Mohtashim.

**Writing – review & editing:** Saumitra Chakravarty, Shah Ishmam Mohtashim.

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
