## [Decision Letter · Decision Letter 0]

16 May 2025

PONE-D-25-13487Mathematical bridge between epidemiological and molecular data on cancer and beyondPLOS ONE

Dear Dr. Mohtashim,

Thank you for submitting your manuscript to PLOS ONE. After careful consideration, we feel that it has merit but does not fully meet PLOS ONE’s publication criteria as it currently stands. Therefore, we invite you to submit a revised version of the manuscript that addresses the points raised during the review process.

We look forward to receiving your revised manuscript.

Kind regards,

Tomasz W. Kaminski

Academic Editor

PLOS ONE

2. We note that your Data Availability Statement is currently as follows: [All relevant data are within the manuscript and its supporting information files.]

3. "Please update your submission to use the PLOS LaTeX template. The template and more information on our requirements for LaTeX submissions can be found at http://journals.plos.org/plosone/s/latex"

5. We note you have included a table to which you do not refer in the text of your manuscript. Please ensure that you refer to Table 1-3 in your text; if accepted, production will need this reference to link the reader to the Table.

Additional Editor Comments:

Dear Authors,

I have now received two reviewers reports for your manuscript, both recommending a major revision. The key concerns raised relate to the need for a clearer introduction and conclusion that align more directly with the study’s objectives and findings, improved clarity and consistency in the presentation of assumptions and mathematical notation, and more detailed explanation of the methodology, model parameters and results. Additional comments addressed language, formatting and the need to update the literature.

I encourage you to address all comments thoroughly in your revised submission.

Best regards,

Tomasz W. Kaminski

Reviewers' comments:

Reviewer's Responses to Questions

**Comments to the Author**

1. Is the manuscript technically sound, and do the data support the conclusions?

Reviewer #1: Partly

Reviewer #2: Partly

2. Has the statistical analysis been performed appropriately and rigorously? 

Reviewer #1: No

Reviewer #2: No

3. Have the authors made all data underlying the findings in their manuscript fully available?

Reviewer #1: Yes

Reviewer #2: No

4. Is the manuscript presented in an intelligible fashion and written in standard English?

Reviewer #1: No

Reviewer #2: No

5. Review Comments to the Author

Reviewer #1: Please correct word processing errors.

I would suggest going through the paper again to catch the odd missing word and some word processing errors, e.g.:

- incorrect list numbering 8. in section 4.2.3, page 11

- please remove paragraph indents after formulas (page 3, after Eq. 1; page 6, after Table 3; page 9, after Eq. 8 and Eq. 9; page 10, after Eq. 11; page 11, after Eq. 17 and Eq. 19)

- lack of explanations of symbols in Eq. 3, Eq. 4, Eq. 6, Eq. 7,

- incorrectly inserted bracket ")" in paragraph 2.2 and uppercase letter "L":

"For the non-linear (convex upwards and concave

upwards model) a simple hybrid algorithm was used: Linear regression algorithm"

- I would suggest using the word "gender" instead of "sex"

- I would suggest writing names in lower case letters in Table 1, Table 2, Table 3 - for example instead of "BLADDER" we can write "Bladder"

- please check the punctuation marks

- the literature is out of date

- are the coefficients of the model statistically significant? Are there any significant differences in the R2 values for the three groups (male, female, both)?

- Section 2.1 "We have tested our model on" - please clarify what we mean by "our model",

- the data set variables are not characterised. There is no description of the interpretation of the models used.

Reviewer #2: The paper “Mathematical bridge between epidemiological and molecular data on cancer and beyond” is a mathematically sound paper, but a lot of improvement is needed. The paper is very mathematical, hard to follow, contains very little interpretation of the models, its parameters and findings, which may reduce the applicability of the proposed models. The Comments Attached.

6. PLOS authors have the option to publish the peer review history of their article (what does this mean?). If published, this will include your full peer review and any attached files.

Reviewer #1: No

Reviewer #2: No

---

## [Author Response · Author response to Decision Letter 1]

15 Jun 2025

We have updated our manuscript according to the comments. The response to reviewers file has been uploaded alongside the updated fresh manuscript and a document showing the changes from the first submission. We hope we have satisfied all the comments.

---

## [Decision Letter · Decision Letter 1]

1 Jul 2025

Mathematical bridge between epidemiological and molecular data on cancer and beyond

PONE-D-25-13487R1

Dear Dr. Mohtashim,

We’re pleased to inform you that your manuscript has been judged scientifically suitable for publication and will be formally accepted for publication once it meets all outstanding technical requirements.

An invoice will be generated when your article is formally accepted. Please note, if your institution has a publishing partnership with PLOS and your article meets the relevant criteria, all or part of your publication costs will be covered. Please make sure your user information is up-to-date by logging into Editorial Manager at Editorial Manager? and clicking the ‘Update My Information' link at the top of the page. If you have any questions relating to publication charges, please contact our Author Billing department directly at authorbilling@plos.org.

Kind regards,

Tomasz W. Kaminski

Academic Editor

PLOS ONE

Reviewers' comments:

Reviewer's Responses to Questions

**Comments to the Author**

1. If the authors have adequately addressed your comments raised in a previous round of review and you feel that this manuscript is now acceptable for publication, you may indicate that here to bypass the “Comments to the Author” section, enter your conflict of interest statement in the “Confidential to Editor” section, and submit your "Accept" recommendation.

Reviewer #2: All comments have been addressed

2. Is the manuscript technically sound, and do the data support the conclusions?

Reviewer #2: Yes

3. Has the statistical analysis been performed appropriately and rigorously?

Reviewer #2: Yes

4. Have the authors made all data underlying the findings in their manuscript fully available?

Reviewer #2: Yes

5. Is the manuscript presented in an intelligible fashion and written in standard English?

Reviewer #2: Yes

6. Review Comments to the Author

Reviewer #2: The authors have done an outstanding job in revising the paper. All the comments have been addressed.

7. PLOS authors have the option to publish the peer review history of their article (what does this mean?). If published, this will include your full peer review and any attached files.

Reviewer #2: No

---

## [Editor Report · Acceptance letter]

PONE-D-25-13487R1

PLOS ONE

Dear Dr. Mohtashim,

I'm pleased to inform you that your manuscript has been deemed suitable for publication in PLOS ONE. Congratulations! Your manuscript is now being handed over to our production team.

Kind regards,

on behalf of

Dr. Tomasz W. Kaminski

Academic Editor

PLOS ONE